# Antinociceptive and Antiallodynic Activity of Some 3-(3-Methylthiophen-2-yl)pyrrolidine-2,5-dione Derivatives in Mouse Models of Tonic and Neuropathic Pain

**DOI:** 10.3390/ijms23074057

**Published:** 2022-04-06

**Authors:** Anna Dziubina, Anna Rapacz, Anna Czopek, Małgorzata Góra, Jolanta Obniska, Krzysztof Kamiński

**Affiliations:** 1Department of Pharmacodynamics, Faculty of Pharmacy, Jagiellonian University Medical College, Medyczna 9 St., 30-688 Krakow, Poland; anna.dziubina@uj.edu.pl; 2Department of Medicinal Chemistry, Faculty of Pharmacy, Jagiellonian University Medical College, Medyczna 9 St., 30-688 Krakow, Poland; malgorzata.gora@doctoral.uj.edu.pl (M.G.); mfobnisk@cyf-kr.edu.pl (J.O.); k.kaminski@uj.edu.pl (K.K.)

**Keywords:** neuropathic pain, inflammatory pain, analgesic activity, anticonvulsants, pirrolidyne-2,5-dione

## Abstract

Antiseizure drugs (ASDs) are commonly used to treat a wide range of nonepileptic conditions, including pain. In this context, the analgesic effect of four pyrrolidine-2,5-dione derivatives (compounds **3**, **4**, **6**, and **9**), with previously confirmed anticonvulsant and preliminary antinociceptive activity, was assessed in established pain models. Consequently, antinociceptive activity was examined in a mouse model of tonic pain (the formalin test). In turn, antiallodynic and antihyperalgesic activity were examined in the oxaliplatin-induced model of peripheral neuropathy as well as in the streptozotocin-induced model of painful diabetic neuropathy in mice. In order to assess potential sedative properties (drug safety evaluation), the influence on locomotor activity was also investigated. As a result, three compounds, namely **3**, **6**, and **9**, demonstrated a significant antinociceptive effect in the formalin-induced model of tonic pain. Furthermore, these substances also revealed antiallodynic properties in the model of oxaliplatin-induced peripheral neuropathy, while compound **3** attenuated tactile allodynia in the model of diabetic streptozotocin-induced peripheral neuropathy. Apart from favorable analgesic properties, the most active compound **3** did not induce any sedative effects at the active dose of 30 mg/kg after intraperitoneal (*i.p.*) injection.

## 1. Introduction

Neuropathic pain is one of the most common chronic diseases that affects 7–10% of the general population [1,2]. It is defined as pain caused by a lesion or disease of the central and peripheral nervous system, resulting from a cascade of neurobiological processes, that leads to hyperexcitability in the direction pathways of somatosensory neuron pathways [3]. Common conditions associated with neuropathic pain include chemotherapy-induced neuropathic pain, diabetic neuropathy, postherpetic, and trigeminal neuralgia [1,4,5].

Due to the fact that classic analgesics such as paracetamol, non-steroidal anti-inflammatory drugs (NSAIDs), or weak opioids, such as codeine, are not effective, current treatment strategies for neuropathic pain are based on anticonvulsant (especially *gabapentinoids*, also known as the α2*δ* ligands, i.e., gabapentin and pregabalin) and antidepressant drugs (serotonin and noradrenaline reuptake inhibitors (SNRIs), particularly duloxetine and tricyclic antidepressants, i.e., amitriptyline). Furthermore, other drugs, including tramadol (with dual mechanisms, namely a weak opioid agonist and serotonin and noradrenaline reuptake inhibitor, topical lidocaine or capsaicin, botulinum toxin A, and ketamine, are used [1,6,7]. Importantly, tapentadol (a strong opioid agonist and noradrenaline reuptake inhibitor) has recently been approved for the treatment of painful diabetic neuropathy [8]. On the other hand, traditional strong opioid agonists, such as morphine and oxycodone, do not provide optimal treatment options due to inadequate tolerability and are, therefore, generally not considered as the first-line analgesic options for neuropathic pain [9].

The analgesic action of antiseizure drugs (ASDs) has been demonstrated in numerous animal studies and clinical trials [1,10,11]. Several gabapentinoids are approved therapies in nonepileptic conditions, including peripheral and central neuropathic pain. Additionally, gabapentin and topiramate are found to be useful in migraine prevention and phenytoin and carbamazepine for the treatment of trigeminal neuralgia [1,12,13].

Our research focused on a group of hybrid molecules (multifunctional ligands) combining the chemical fragments of well-established ASDs as compounds with potential anticonvulsant and antinociceptive properties [11]. It should be stressed that multimodal antiseizure medications (i.e., valproic acid, cenobamate, etc.) are known to be more effective in the control of epileptic seizures compared with single-target drugs (i.e., lamotrigine, lacosamide, etc.) [14]. Other preclinical studies on the hybrid compounds for the treatment of neuropathic pain are also reported [15,16,17].

The most recent studies reported by our team revealed potent anticonvulsant activity for several new hybrid compounds with pyrrolidine-2,5-dione and thiophene rings in the structure [11,18,19]. These compounds were effective in the maximal electroshock seizure (MES) test (which is a model of tonic–clonic seizures in humans) and the 6 Hz psychomotor seizure test (recognized as a model of focal seizures in humans) [18]. Furthermore, the selected anticonvulsants identified previously, that is, **3**, **4**, **6**, and **9** (original numbering from Góra et al. [18]), also revealed peripheral analgesic activity in the writhing test (a model of inflammatory pain), and two of them (**6**, **9**) were additionally active in the hot plate test (a model of acute pain). The structures of compounds **3**, **4**, **6**, and **9** are shown in Figure 1.

Our research, presented in this article, provides a promising avenue for the development of novel and original hybrid analgesics. Taking into account potent anticonvulsant activity and the promising analgesic properties of compounds **3**, **4**, **6**, and **9**, the aim of the current study was to perform a more detailed analysis of their analgesic activity, with a particular emphasis on pharmacoresistant chronic neuropathic pain models. Therefore, we examined the antinociceptive activity of these compounds in a tonic pain model (formalin test) and antiallodynic/antihyperalgesic activity in the oxaliplatin-induced neuropathic pain model of chemotherapy-induced peripheral neuropathy, as well as in the streptozotocin-induced model of painful diabetic neuropathy in mice administered by intraperitoneal injection (*i.p.*). For the preliminary estimation of the behavioral safety (potential sedative properties), the influence on the locomotor activity of mice was also studied.

## 2. Results

### 2.1. Chemistry

The chemical synthesis of 3-(3-methyltiophen-2-yl)-pyrrolidine-2,5-diones (**3**, **4**, **6**, **9**) is depicted in Figure 1. As previously reported [18], these compounds were obtained in a two-step synthesis. In the first step, the 2-(3-methylthiophen-2-yl)succinic acid (**I**) was prepared using the Abeijon et al. method [20] and then condensed with aminoalkylmorpholine, to form the final compounds **3** and **4,** or with 1-(3-(trifluoromethyl)phenyl)piperazine/1 (3,4 dichlorophenyl)piperazine to form compounds **6** and **9**, respectively.

### 2.2. Antinociceptive Activity in the Formalin Test

The formalin test was used as a tonic model of nociception. In the first (neurogenic) phase of the test, **3** and **4,** formed at doses of 30 and 45 mg/kg and 30 and 60 mg/kg, respectively, did not attenuate the nociceptive response. In the same phase, **6,** at doses 30 and 45 mg/kg, showed a prominent antinociceptive effect, as it significantly decreased the duration of the licking response by 61% (*p* < 0.001) and 80% (*p* < 0.001), respectively. Compound **9,** at doses of 30, 45, and 60 mg/kg, also significantly reduced the pain: by 23.6% (*p* < 0.05), by 51.7% (*p* < 0.001), and by 34% (*p <* 0.01), respectively (Figure 2A).

In the second (late) phase of the formalin test, only compound **4** did not reduce the duration of the licking response. For the other three compounds, a statistically significant reduction in pain response was observed as follows: compound **3** at doses of 30 and 45 mg/kg: 42% (*p* < 0.01) and 55% (*p* < 0.001); compound **6** at doses of 30 and 45 mg/kg: 89.6% (*p* < 0.001) and 97% (*p* < 0.001); compound **9** at doses of 30, 45, and 60 mg/kg: 32.4% (*p* < 0.05), 50.7% (*p <* 0.01), and 37% (*p* < 0.05) (Figure 2B).

Previous studies have shown that the reference drug, pregabalin, tested at doses of 1, 10, and 30 mg/kg, reduced the duration of the pain response in the second phase of the test by 37.8% (*p* < 0.05), 59.3% (*p* < 0.01), and 73.7% (*p* < 0.001), respectively [21].

### 2.3. Antiallodynic Activity in Oxaliplatin-Induced Neuropathic Pain

Tactile allodynia was measured in the oxaliplatin-treated mice 3 h (acute allodynia) and 7 days (late allodynia) after injection. The mean force that caused paw withdrawal was 2.85 ± 0.04 g in the control group (animals not treated with oxaliplatin), while this value ranged from 1.69 ± 0.08 g to 2.06 ± 0.04 g in the group of animals treated with oxaliplatin after 3 h and from 1.87 ± 0.03 g to 2.09 ± 0.04 g after 7 days, respectively (*p* < 0.0001 in all groups). A total of 3 h after oxaliplatin administration, compounds **3**, **6**, **9** at the dose of 30 mg/kg significantly increased the pain threshold by 15% (*p <* 0.01), by 28% (*p* < 0.001), and by 35% (*p* < 0.001), respectively. The most effective was compound **9,** at a dose of 45 mg/kg, since it elevated the pain sensitivity threshold by 47% in the acute phase of the test. Pregabalin, used as reference drug, at doses of 10 mg/kg and 30 mg/kg increased the pain sensitivity threshold by 51% (*p* < 0.001) and 63% (*p* < 0.001), respectively. The results are presented in Figure 3A.

A weaker effect of reducing allodynia was observed in the late phase of allodynia. Among the compounds studied, only **3** and **6,** at a dose of 30 mg/kg, raised the pain threshold by 16.0% and 20%, respectively. Compound **9,** at the same dose, did not affect the nociceptive threshold. However, at a higher dose of 45 mg/kg, tactile allodynia reduced significantly by 25%. This effect was comparable to that of pregabalin at a dose of 10 mg/kg, which raised the pain threshold by 28%. In turn, at a dose of 30 mg/kg, pregabalin reduced allodynia by 48%. The results are presented in Figure 3B.

### 2.4. Influence on Tactile Allodynia (Von Frey Test) and Heat Hyperalgesia (Hot Plate Test)

In the von Frey test in a non-diabetic control group, the mean pain sensitivity threshold for mechanical stimulation was 2.95 ± 0.03 g, while in streptozotocin-treated mice, sensitivity to pain increased significantly, resulting in a reduction of the mechanical nociceptive threshold to a value range from 1.89 ± 0.04 g to 1.81 ± 0.03 g. In the von Frey test, in streptozotocin-treated mice, compound **3** significantly increased the pain thresholds at the dose of 30 mg/kg of 1.89 ± 0.04 to 2.10 ± 0.04 g (*p* < 0.01), and at a dose of 45 mg/kg from 1.81 ± 0.03 to 2.58 ± 0.03 s. In this model, **3** at the doses of 30 and 45 mg/kg statistically significantly attenuated allodynia by 12% and 43%, respectively.

In the hot plate test, in a normoglycemic control group, the baseline latency to the pain reaction was 30.24 ± 1.8 s, while in streptozotocin-treated mice the values obtained were lower, ranging from 16.02 ± 1.93 s to 16.06 ± 2.67 s, and were statistically significant (*p* < 0.001). In the streptozotocin-treated mice, compound **3** only slightly prolonged the latency time to the pain reaction at a dose of 30 mg/kg from 16.02 ± 1.93 to 24.71 ± 2.42 s, and at a dose of 45 mg/kg from 16.06 ± 2.67 to 22.81 ± 2.46 s. As in the statistically significant allodynic effect after streptozotocin injection observed in the von Frey test, herein we observe the statistically significant development of heat hyperalgesia. The results are presented in Figure 4A,B. Previously, pregabalin tested in our laboratory under the same conditions has been reported to show antihyperalgesic activity at doses of 1–30 mg/kg, and antiallodynic activity at doses of 10 and 30 mg/kg [22].

### 2.5. Influence on Locomotor Activity

A significant influence on the locomotor activity of mice was recorded for compounds **6** and **9** at doses of 30 mg/kg (*p* < 0.001) and 45 mg/kg (*p* < 0.001), respectively. For compounds **3**, **4**, and **9,** at a dose of 30 mg/kg, no significant influence on locomotor activity was observed. The results are presented in Figure 5. Furthermore, pregabalin, at a dose of 30 mg/kg, did not significantly influence mice’s locomotor activity [21].

## 3. Discussion

Neuropathic pain is caused by damage or disease of the nervous system, particularly the somatosensory part of it. It is a multifactorial disorder with many different mechanisms, i.e., neuroinflammation, oxidative stress, altered ion channel function, and axonal degeneration [23,24]. Neurological disorders such as epilepsy and neuropathic pain have similar pathophysiology, allowing some ASDs to be useful for the treatment of several neuropathic pain conditions [25]. ASDs with different mechanisms have been shown to elicit analgesic effects, particularly in reducing neuropathic pain, in many animal studies and clinical trials [1,10,11].

The development of new hybrid compounds for the treatment of neuropathic pain treatment is constantly emerging, especially due to the expected higher efficacy of such molecules in the pharmacotherapy of this complex disease [11,19,26,27]. Therefore, our research was focused on a group of hybrid compounds characterized by a multimodal mechanism of action and the ability to alleviate neuropathic pain caused by various factors, i.e., chemotherapy and diabetes. These compounds were designed as multifunctional ligands containing key structural fragments of two well-known ASDs, namely tiagabine and ethosuximide, that is, 3-methylthiophene and pyrrolidine-2,5-dione, respectively, which are responsible for efficacy or pharmacokinetic properties. They also possess an interesting, multi-target mechanism of action, namely the blocking of neuronal voltage-sensitive sodium channels (site2), L-type calcium channels, and TRPV1 receptors. Although hybrid compounds have been previously characterized in two acute tests of analgesic activity (writhing and hot plate tests) [18], they have not been characterized in the tonic pain model and in the models of two common forms of chronic neuropathic pain, i.e., chemotherapy-induced peripheral neuropathy and the streptozotocin-induced model of painful diabetic neuropathy. In the current study, we present a detailed analysis of analgesic activity, with a particular emphasis on pharmacoresistant chronic neuropathic pain. For this purpose, we implemented routinely used and widely recognized mouse models of tonic and neuropathic pain to assess the analgesic properties of the aforementioned hybrid compounds.

The results reported herein are a continuation of our research in the group of 3-(3-methylthiophen-2-yl)pyrrolidine-2,5-dione derivatives, with previously confirmed anticonvulsant and analgesic activity [18]. Similar studies for chemically diversified substances were previously reported by our team [21,22,28,29,30,31,32,33,34].

Based on the results obtained, in the current studies, the most active compounds **3**, **4**, **6**, and **9** characterized by beneficial drug-likeness, potent anticonvulsant activity in MES and 6 Hz seizure models, as well as peripheral analgesic efficacy in the writhing test, were tested in more detailed pain models.

In the first step, the formalin test was used to evaluate the efficacy of selected compounds **3**, **4**, **6**, and **9**. Importantly, this animal model of pain showed the activity then observed in a clinical trial of drugs in the treatment of persistent tonic pain [35]. Formalin-induced pain involves numerous mechanisms, such as mediators, channels, and receptors [36,37,38], as well as nerve fiber types [39] and signaling pathways [38] involved in formalin-induced inflammatory nociception. Formalin behaviors consist of an early phase (0–5 min, phase I) followed by brief quiescence and a late phase (15–30 min, phase II). The acute neurogenic/chemical phase (phase I) is dependent on the activation of sensory C-fibers and is mediated by the activation of the transient receptor potential cation channel (TRPA1) [40]. The acute phase is followed by a continuous and longer-lasting tonic phase II mediated by either a central sensitization state or the inflammation-induced hyperactivity of the afferent nociceptors, or a combination of both. The late phase of formalin represents persistent and tonic pain [41]. The available data show the potential effectiveness of some ASDs in the formalin model of pain [10,41]. They are effective in both phases of the test as lacosamide [28,29], tiagabine [30], and valproic acid [31], or only in the second phase of the test as pregabalin [21], lamotrigine [41,42,43], and gabapentin [42,43].

In this test, in the first neurogenic phase, two compounds, **6** at doses of 30 and 45 mg/kg and **9** at doses of 30, 45, and 60 mg/kg, significantly attenuated the pain response. The other two compounds, **3** at 30 mg/kg and 45 mg/kg and **4** at 30 mg/kg and 60 mg/kg, were inactive in this phase. In the second inflammatory phase of the test, the three compounds **3**, **6**, and **9**, strongly and significantly reduced pain. Only compound **4** did not reveal any significant analgesic activity in this test, proving the lack of an analgesic effect.

In the next step, compounds showing statistically significant analgesic activity in the second inflammatory phase of the formalin test were tested in an animal model of oxaliplatin-induced peripheral neuropathy. This model is used to identify compounds with antiallodynic activity and to elucidate the mechanism of neuropathy. Oxaliplatin is a third-generation platinum-based chemotherapeutic drug that is widely used to treat various types of cancer. Several diverse mechanisms of the toxic effects of this drug have been described [44]. Sodium, potassium, and calcium ion channels and different types of transient receptor potential families (TRPA1, TRPM8, and TRPV1) were shown to be involved, as their function altered after oxaliplatin injection [5,45]. Furthermore, increased activity of spinal glial cells and pro-inflammatory and neuroexcitatory cytokines in the dorsal horn of the spinal cord were associated with oxaliplatin [5]. ASDs act at several sites that may be relevant to neuropathic pain, but the precise mechanism of their effect remains unclear. Relevant sites of action include voltage-gated ion channels (sodium and calcium), ligand-gated ion channels transient receptor TRPV1, and others [46]. Furthermore, ASDs that block sodium channels, and high-frequency action potential firing, are effective in the treatment of neuropathic pain and seizures [28,32,47]. Taking this into account, three compounds **3**, **6**, and **9,** for which affinity for calcium, sodium, and TRPV1 channels had been previously confirmed [18], were examined to establish their antiallodynic efficacy in the oxaliplatin-induced neuropathic pain model. Administration of oxaliplatin causes painful neuropathy in animals and humans, with symptoms such as cold and mechanical allodynia [48]. ASDs, such as gabapentin and pregabalin, significantly inhibit allodynia induced by oxaliplatin, paclitaxel, and vincristine [49,50]. The previous data demonstrated that the administration of dual TRPA1 and phosphodiesterase antagonists also reduces oxaliplatin-induced tactile allodynia in mice [51].

In the present studies, all compounds and the reference drug (pregabalin) increased the pain threshold statistically significantly, i.e., attenuated tactile allodynia. In the acute phase of neuropathy, all compounds tested at a dose of 30 mg/kg showed a marked elevation of the mechanical nociceptive threshold. Additionally, in the late phase of allodynia, compounds **3** and **6** at a dose of 30 mg/kg decreased tactile allodynia weaker than in the first phase, but significantly, while compound **9** did not reveal any significant antiallodynic activity. The same compound, at a higher dose of 45 mg/kg, statistically significantly reduced allodynia in both phases. Pregabalin, as a reference drug, tested at doses 10 and 30 mg/kg in both phases of the test significantly, and in a dose-dependent manner, reduced allodynia, with the strongest effect observed at a dose of 30 mg/kg. The findings presented here indicate that the studied compounds are more effective in attenuating acute versus persistent symptoms of chemotherapy-induced neuropathic pain.

Neuropathic pain is also one of the most common complications associated with diabetes. Approximately 30–50% diabetic neuropathy patients develop neuropathic pain, which most commonly takes the form of a spontaneous burning pain of the feet. In the treatment of diabetic neuropathic pain, anticonvulsants and antidepressant drugs (SNRIs and tricyclic antidepressants) are used [52]. A streptozotocin-induced diabetic animal model has been widely used to study the mechanism of diabetic neuropathic pain and evaluate potential therapies [53]. In this model, depending on the time course of the disease, the animals show either tactile allodynia and thermal hyperalgesia (2–8 weeks of diabetes) or thermal hypoalgesia (in the advanced stage, ≥12 weeks of diabetes) [54]. In the present study, after 21 days, diabetic mice exhibited significantly increased blood glucose levels [55], urine output, and decreased body weight gain (data not shown) compared with their normoglycemic group. Under the protocol used in the present study, diabetic mice developed a marked decrease in mechanical and thermal nociceptive thresholds that result in the development of tactile allodynia and heat hyperalgesia, respectively.

In these studies, compound **3**, which did not reveal cytotoxic properties or elicit any significant change in locomotor activity in mice at the doses tested, was examined [18]. Under these conditions, compound **3** exhibited a significant and dose-dependent reduction in tactile allodynia in diabetic neuropathic mice. In previous studies, lacosamide had a similar effect [21].

A less pronounced effect of the compound tested on thermal hyperalgesia was observed. In other words, a single dose (30 and 45 mg/kg) of compound **3** only slightly influenced the pain responses of diabetic mice in the hot plate test. On the contrary, unlike the compound investigated, antiseizure drugs, such as pregabalin [22], lacosamide [21,29], and levetiracetam [56], significantly attenuated thermal hyperalgesia. This effect is likely because compound **3** did not show activity in the hot plate assay in the normoglycemic mice at these doses. Furthermore, the low affinity for the TRPV1 receptor and/or sodium channel may suggest a weak effect on reducing thermal hyperalgesia, but the final explanation of these data remains to be clarified. In summary, although compound **3** at doses 30 and 45 mg/kg can significantly mitigate the signs of tactile allodynia, it attenuates thermal hyperalgesia to a much lesser extent. It seems plausible that the weaker analgesic effect of the tested compound could hinder its potential use in attenuating pain in diabetic neuropathy.

An important aspect of the study is also to check whether the new compounds in the active doses affect spontaneous locomotor activity. We tested the influence of compounds on locomotor activity to eliminate the possibility of false positive results and ambiguous interpretation of the in vivo results, which could be due to the sedative properties of the compounds. The data obtained indicate that only **6,** at the dose of 30 mg/kg, and **9,** at the highest dose of 45 mg/kg, decreased locomotor in mice. Compounds **3**, **4,** and **9,** at an analgesic-active dose of 30 mg/kg, did not influence the locomotor activity of mice. Drug-induced sedation can significantly limit the use of treatment but does not exclude its potential use in pain management. Sedative properties of treatments may be desirable in patients suffering from insomnia resulting from pain sensations [57].

## 4. Materials and Methods

### 4.1. Chemistry

The precise synthetic procedure to obtain the final compounds (**3**): 3-(3-methylthiophen-2-yl)-1-(2-morpholinoethyl)pyrrolidine-2,5-dione hydrochloride, (**4**): 3-(3-methylthiophen-2-yl)-1-(3-morpholinopropyl)pyrrolidine-2,5-dione hydrochloride, (**6**): 3-(3-methylthiophen-2-yl)-1-(3-(4-(3-(trifluoromethyl)phenyl)piperazin-1-yl)propyl)pyrrolidine-2,5-dione hydrochloride, and (**9**): 1-(3-(4-(3,4-dichlorophenyl)piperazin-1-yl)propyl)-3-(3-methylthiophen-2-yl)pyrrolidine-2,5-dione hydrochloride, as well as detailed analytical data, were previously published in Góra et al. [18].

### 4.2. Pharmacology

#### 4.2.1. Animals

The experiments were carried out on male CD-1 mice (18–26 g) provided by an accredited animal facility (Jagiellonian University Medical College, Krakow, Poland). Animals were housed in groups of 10 in standard plastic cages, at a room temperature of 20 ± 2 °C, exposed to a 12:12 h light/dark cycle, with ad libitum food and water. All experiments were performed between 9 a.m. and 3 p.m. The tested groups, consisting of 8–10 mice, were chosen by means of a randomized schedule. Care was taken to minimize animal suffering and reduce the number of animals used (3R policy).

#### 4.2.2. Drug Administration

All substances were suspended in a 1% Tween 80 solution and administered intraperitoneally (*i.p*.) in a volume of 10 mL/kg of body weight 30 min prior to the test. Doses of compounds **3**, **4**, **6**, and **9** were chosen based on previous studies (in vivo test) presented by Góra et al., 2020 [18].

#### 4.2.3. Tonic Pain Model: The Formalin Test

The procedure was based on the method described by Hunskaar and Hole [58] with some modifications by [41]. Briefly, 20 μL of 2.5% formalin solution was injected, intraplantar (*i.pl.*), into the right hind paw of the mouse. Immediately after injection, the animals were individually placed into a glass beaker and observed for the next 30 min. The time (in seconds) spent licking or biting the hind paw injected at selected intervals, 0–5 and 15–30 min, was measured in each experimental group and was an indicator of nociceptive behavior. The number of pain-evoked responses was summed for all 5-min bins for each mouse and then averaged for each treatment group.

#### 4.2.4. Assessment of Tactile Allodynia in Oxaliplatin-Induced Neuropathic Pain: The Von Frey Test

Oxaliplatin was administered as a single injection of *i.p* at a dose of 10 mg/kg to induce peripheral neuropathy, as previously described by Sałat et al. [34] and Rapacz et al. [21]. Antiallodynic activity was assessed 3 h (acute allodynia) and 7 days (late allodynia) after oxaliplatin administration. Mechanical hyperalgesia was tested, as previously described in detail [34], using the electronic von Frey device (Panlab, Spain). On the day of the experiment (the 1st and 7th day), after a 30 min habituation period, each mouse was tested three times in the plantar region of the hind paw, to obtain baseline values. Subsequently, the mice were pretreated with the tested compounds and the reference drug, pregabalin. A total of 30 min later, the animals were tested again, and the mean values of the mechanical withdrawal threshold were obtained for each mouse. The reference drug, pregabalin, was used in accordance with the previous method [21].

#### 4.2.5. Streptozotocin-Induced Diabetic Neuropathy

##### Induction of Diabetes

To induce diabetes, mice were injected with a single *i.p.* dose of streptozotocin (200 mg/kg), dissolved in a 0.1 N citrate buffer. The body weight of the mice was monitored before and after streptozotocin injection. Blood glucose levels were measured 1, 2, and 3 weeks after streptozotocin injection using a blood glucose monitoring system (AccuChek Active, Roche, France). Blood samples (5 µL) for the glucose concentration measurement were obtained from the mouse tail vein. The animals were defined as diabetic when their blood glucose concentration exceeded 300 mg/dl and only these mice were used as diabetic mice in further tests, following the protocol described previously [22,55].

##### Assessment of Tactile Allodynia in Streptozotocin-Treated Mice: The Von Frey Test

Mechanical hypersensitivity (tactile allodynia) was assessed using the electronic von Frey unit (Panlab, Cornellà de Llobregat, Spain), as described above in the section “Oxaliplatin-induced neuropathic pain model” [21,34]. The test was performed 21 days after streptozotocin administration.

##### Assessment of Heat Hyperalgesia in Streptozotocin-Treated Mice: Hot Plate Test

Thermal hyperalgesia was assessed in the hot plate test using the hot/cold plate apparatus (Panlab/Harvard Apparatus, Cornellà de Llobregat, Spain) as described by Eddy and Leimbach [59]. In this test, after the establishing of pre-drug latency to pain reaction for each animal, the mice were treated with tested compounds and, 30 min later, placed on a hot plate set at 55 °C and observed for a nocifensive response (hind paw licking or jumping). The cut-off time was established at 45 s.

#### 4.2.6. Spontaneous Locomotor Activity

Spontaneous locomotor activity was monitored using actometers (40 × 40 × 31 cm) (Ugo Basile, Gemonio, Italy) as previously described [33]. Thirty minutes after the injection of the investigated compounds, mice were placed individually in the actometers for 30 min of habituation. After that, the number of light-beam crossings (number of movements) for a 30-min session was counted automatically.

#### 4.2.7. Data Analysis

The data were analyzed using GraphPad Prism software. Data are expressed as mean ± SD. Statistically significant differences between groups were calculated using one-way analysis of variance (ANOVA) and the post hoc Dunnett multiple comparison test, the post hoc Bonferroni multiple comparison test, or the post hoc Sidak multiple comparison test when appropriate. The significance criterion was established at *p* < 0.05.

## 5. Conclusions and Perspectives

In the present study, the antinociceptive properties of new 3-(3-methylthiophen-2-yl)pyrrolidine-2,5-dione derivatives, with previously confirmed anticonvulsant and analgesic activity, were evaluated in the model of tonic pain, as well as in models of chemotherapy- and diabetic-induced peripheral neuropathy. Among the compounds studied, only compound **4** did not show any analgesic activity in the formalin test. The other compounds **6** and **9** demonstrated a significant antinociceptive effect in both phases of the test, whereas compound **3** was effective only in the second phase of the formalin test. Compounds **3**, **6,** and **9** revealed antiallodynic properties in the acute and late phases of oxaliplatin-induced neuropathy. Furthermore, compound **3** was able to reduce tactile allodynia but not thermal allodynia in the mouse model of streptozotocin-induced painful diabetic nephropathy. At an active dose of 30 mg/kg, only compound **6** diminished spontaneous animal movements. In conclusion, the results reported in the current paper confirm that anticonvulsants, characterized by multi-target mechanism of action, namely, the blockade of the TRPV1 receptor, voltage-gated sodium, and/or L-calcium channels, may assure an interesting and non-addictive option for the treatment of various pain conditions.

The results obtained herein supplement the knowledge on the development of new therapeutics for the control of neuropathic pain among compounds with the hybrid structure. These multimodal compounds proved to be effective in different acute and chronic pain models that creates hope for the identification of new, broad-spectrum, and nonopioid analgesics in the near future.

## Data Availability

Not applicable.

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
