# Peer review of "Antinociceptive and Antiallodynic Activity of Some 3-(3-Methylthiophen-2-yl)pyrrolidine-2,5-dione Derivatives in Mouse Models of Tonic and Neuropathic Pain"

_ijms, 2022, doi:10.3390/ijms23074057_

Round 1

Reviewer 1 Report

This is an interesting work presenting antinociceptive and antiallodynic activity of novel antiseizure substances. Occasionally English language should be corrected, but generally manuscript is well written.

Minor comments:

Line 15: choose one: “especially” or “including”

Lines 18-21: please rephrase a sentence, it is too long

Line 26: compound 3

Line 55: abbreviation for ADS?

Line 57-59: please rephrase a sentence

Lines 111-113 and lines 128-130: you mentioned pregabalin used as a reference drug, but you do not mention this in the Methods section, so please mention it also there.

Line 116: results here and everywhere throughout the manuscript are presented as mean ± SEM, the data are more precisely summarized with SD, please change this

Line 118: I guess that these statistically significant results are related to the comparison with the vehicle?

Discussion: I think it is generally better to start with the discussion of the results and not to make a long literature review of the already known facts; so maybe you could erase the lines 188-195

Line 329-330: what was the dose and how did you decide which dose to administer?

Line 343: the substances were applied again 7 days later?

Line 379: why are data not expressed as mean ± SD? Did you also check the normality of data?

Lines 381-383: how did you decide which post hoc test to use?

Author Response

In response to Reviewer 1, we want to underline that all your comments have been taken into consideration. We appreciate your detailed and professional comments that have helped us to improve our manuscript. The following changes have been introduced to our manuscript:

  • Line 15: choose one: “especially” or “including”

Following the Reviewer’s suggestion one word (including) was chosen.

  • Lines 18-21: please rephrase a sentence, it is too long

The sentence was rephrased in line with the Reviewer’s suggestion.

  • Line 26: compound 3

The word “compound” was added.

  • Line 55: abbreviation for ADS?

The acronym ASDs was expanded (Antiseizure drugs).

  • Line 57-59: please rephrase a sentence

The sentence was corrected.

  • Lines 111-113 and lines 128-130: you mentioned pregabalin used as a reference drug, but you do not mention this in the Methods section, so please mention it also there.

Pregabalin was added in the Methods section, as was suggested by the Reviewer.

  • Line 116: results here and everywhere throughout the manuscript are presented as mean ± SEM, the data are more precisely summarized with SD, please change this

Results were presented as mean ± SD, throughout the manuscript.

  • Line 118: I guess that these statistically significant results are related to the comparison with the vehicle?

The statistically significant results are related to the vehicle-treated mice group.

  • Discussion: I think it is generally better to start with the discussion of the results and not to make a long literature review of the already known facts; so maybe you could erase the lines 188-195

Lines 188-195 were deleted.

  • Line 329-330: what was the dose and how did you decide which dose to administer?

The doses of compounds 3, 4, 6, and 9 were chosen based on previous studies (in vivo test) presented by Góra et al.,2020 [15].

  • Line 343: the substances were applied again 7 days later?

A single dose of oxaliplatin was administered. A suitable description was added to the manuscript: Tested compounds were administered again 7 days later. “On the day of the experiment (the 1st and 7th day),…

  • Line 379: why are data not expressed as mean ± SD? Did you also check the normality of data?

The normality of the data was checked. The results were presented as mean ± SEM as in the publication by Góra et. al 2020 as well as by Paudel et al., 2011.

  • Lines 381-383: how did you decide which post hoc test to use?

The post-hoc test was selected based on statistical analysis performed with GraphPad Prism 8. A positive test result (after a rejection of the null hypothesis) does not answer the question of which expected values differ from each other. Such an answer is provided by tests after the analysis of variance, called post-hoc comparisons. The choice of the post hoc test depends on the comparisons we intend to make, e.g. if we compare groups (more than 2) with controls we can use Dunnet's test (formalin test). We perform multiple comparison tests when, on the basis of the analysis of variance, we find that a factor significantly affects the trait under study. We use them to make pairwise comparisons of means in all combinations (Bonferroni's and Sidak’ tests).

Reviewer 2 Report

Dear Authors,

The present study evaluates the antinociceptive and antiallodynic activity of new 3-(3-methyl-thiophen-2-yl)pyrrolidine-2,5-dione derivatives in mouse models of tonic and neuropathic pain. The research subject is interesting and brings scientific important data in the field, as it deals with a subject that is currently of great interest. Some changes of the manuscript should nevertheless be performed in order to improve its quality. Following specific changes should thus be performed:

 Major changes

Title: As your compounds are previously described in other studies, I think the word “new” should not be in the title.

Abstract: It is not clear what numbers 3, 4, 6, 9 represent. Please rephrase lines 14-17 and clarify “Considering 21 the drug safety evaluation,”.

Introduction:  This section should contain information regarding similar existing studies in literature and, in comparison, authors should emphasize the novelty and originality of their study. Please explain. You only cite your previous studies – are they the only ones on the same subject? Please give more details on them and add informations on similar existing studies, in order to be able highlight novelty and originality compared to them. If these studies do not exist, mention. At the same time, the aim of the study is not clear. The last paragraph should contain it, but it does not and it is necessary to add it. Please modify accordingly.

Discussions: Novelty and originality of the present study should once again be emphasized here, by comparison with similar studies of the results that are obtained in the present study. This should be performed especially if similar studies exist. If not, please state and highlight your novelty. You cite your previous studies, but are they singular? If so, please mention and emphasize novelty by comparing with them also. You have comments on your results but it is not enough.

Materials and methods: Some of the cited methods do not have references. Are the methods described hereby completely new? If not and they were at least adapted, please add references where necessary.

Conclusions: Please give perspectives of your study.

Please revise style of references. It does not respect the recommendations of the journal.

All these suggested changes should be performed in order to bring further improvements to the manuscript.

Author Response

In response to Reviewer 2, we wanted to underline that all your comments have been taken into consideration. We appreciate your detailed and professional comments that have helped us to improve our manuscript. The following changes have been introduced to our manuscript:

The present study evaluates the antinociceptive and antiallodynic activity of new 3-(3-methyl-thiophen-2-yl)pyrrolidine-2,5-dione derivatives in mouse models of tonic and neuropathic pain. The research subject is interesting and brings scientific important data in the field, as it deals with a subject that is currently of great interest. Some changes of the manuscript should nevertheless be performed in order to improve its quality. Following specific changes should thus be performed:

  • Title: As your compounds are previously described in other studies, I think the word “new” should not be in the title.

The word “new” has been deleted.

  • Abstract: It is not clear what numbers 3, 4, 6, 9 represent. Please rephrase lines 14-17 and clarify “Considering 21 the drug safety evaluation,”.

The sentence was rephrased in line with the Reviewer’s suggestion.

  • Introduction: This section should contain information regarding similar existing studies in literature and, in comparison, authors should emphasize the novelty and originality of their study. Please explain. You only cite your previous studies – are they the only ones on the same subject? Please give more details on them and add information on similar existing studies, in order to be able highlight novelty and originality compared to them. If these studies do not exist, mention. At the same time, the aim of the study is not clear. The last paragraph should contain it, but it does not and it is necessary to add it. Please modify accordingly.

Following the Reviewer’s suggestion, the necessary citations and comments were added. The purpose of the study was clarified. Please, see the Introduction section where all changes were highlighted in yellow.

  • Discussions: Novelty and originality of the present study should once again be emphasized here, by comparison with similar studies of the results that are obtained in the present study. This should be performed especially if similar studies exist. If not, please state and highlight your novelty. You cite your previous studies, but are they singular? If so, please mention and emphasize novelty by comparing with them also. You have comments on your results but it is not enough.

In the Discussion section, the novelty and originality of the study were underlined once again as suggested by the Reviewer. We have added an entire paragraph to the beginning of the section, please see any changes that have been highlighted in yellow.

  • Materials and methods: Some of the cited methods do not have references. Are the methods described hereby completely new? If not and they were at least adapted, please add references where necessary.

The methods were partially adapted; the references were added.

  • Conclusions: Please give perspectives of your study. Please revise style of references. It does not respect the recommendations of the journal. All these suggested changes should be performed in order to bring further improvements to the manuscript.

The perspectives of the study were added. All references were revised.

Round 2

Reviewer 2 Report

Dear Authors,

The present study evaluates the antinociceptive and antiallodynic activity of new 3-(3-methyl-thiophen-2-yl)pyrrolidine-2,5-dione derivatives in mouse models of tonic and neuropathic pain. The authors performed most of the suggested changes after the first round of review. However, following specific changes should still be performed:

 Major changes

Title suggestion: You may add the word “some” to suggest you test different derivatives.

Abstract: It is not clear what numbers 3, 4, 6, 9 represent. It is also not clarified what does “drug safety evaluation” mean.

Introduction:  This section has not suffered the previously suggested changes. It should contain information regarding similar existing studies in literature and, in comparison, authors should emphasize the novelty and originality of their study. Are your previous studies the only ones on the same subject? Please add informations on similar existing studies if they exist, in order to be able highlight novelty and originality compared to them. If these studies do not exist, mention.

All these suggested changes should be performed in order to bring further improvements to the manuscript.

Author Response

In response to Reviewer 2, we want to underline that all your comments have been taken into consideration. We appreciate your detailed and professional comments that have helped us to improve our manuscript. The following changes have been introduced to our manuscript: 

  • Title suggestion: You may add the word “some” to suggest you test different derivatives.

Following the Reviewer’s suggestion the word “some” has been added.

  • Abstract: It is not clear what numbers 3, 4, 6, 9 represent. It is also not clarified what does “drug safety evaluation” mean.

The numbers 3, 4, 6, 9 are the numbers of the test compounds. For the sake of clarity the word “compounds” has been added.

In our research, the expression "drug safety evaluation" refers to testing new compounds that may have sedative properties. For this purpose, the effect on spontaneous motor activity was measured. The tested compound, which inhibits the locomotor activity of mice, has a sedative effect compared to the control. Drug – induced sedation can hinder the use of compounds as an analgesic and / or anticonvulsant drug. Moreover, in the preclinical study, sedation can influence antinociceptive activity and as a result false positive scores can be obtained.

  • Introduction:  This section has not suffered the previously suggested changes. It should contain information regarding similar existing studies in literature and, in comparison, authors should emphasize the novelty and originality of their study. Are your previous studies the only ones on the same subject? Please add informations on similar existing studies if they exist, in order to be able highlight novelty and originality compared to them. If these studies do not exist, mention.

Following the Reviewer suggestion, the necessary citation [15-17] and comments on similar existing studies were added. Our research, presented in this article, provides a promising avenue for the development of novel and original hybrid analgesics.

 We trust that the above explanation will meet your requirements.